# Corticosterone Injection Impairs Follicular Development, Ovulation and Steroidogenesis Capacity in Mice Ovary

**DOI:** 10.3390/ani9121047

**Published:** 2019-11-29

**Authors:** Yinghui Wei, Weijian Li, Xueqing Meng, Liangliang Zhang, Ming Shen, Honglin Liu

**Affiliations:** College of Animal Science and Technology, Nanjing Agricultural University, Nanjing 210095, China; wyh2017105016@njau.edu.cn (Y.W.); 2017105018@njau.edu.cn (W.L.); 2017105017@njau.edu.cn (X.M.); 2018105016@njau.edu.cn (L.Z.)

**Keywords:** corticosterone, stress, ovulation, follicle development, steroid hormones

## Abstract

**Simple Summary:**

Researchers have hitherto established hundreds of animal stress models. However, these models have some limitations due to the complexity in operation and large differences between individual animals. In particular, there are few stress models that are specifically applied in mammalian ovaries. In this study, using intraperitoneal injection of cortisol/corticosterone (CORT), we successfully established a stress model that acts on the ovarian function. Our data showed that CORT inhibits ovarian and follicular development and blocks ovulation. The establishment of this model might provide a living platform for studying ovarian stress in future research.

**Abstract:**

The aim of this study is to establish an ovarian stress model, and to investigate the effects of stress on follicular development. Our data showed that continuous intraperitoneal injection of CORT successfully created a stressful environment in the ovary. To assess the effects of CORT on ovarian functions, 80 three-week-old ICR (Institute of Cancer Research) female mice were randomly divided into control group and treatment group. All mice were injected intraperitoneally with pregnant horse serum gonadotropin (PMSG). At the same time, the treatment group were injected with CORT (1 mg/mouse) at intervals of 8 h; while the control group was injected with same volume of methyl sulfoxide (DMSO). Blood, ovaries, or ovarian granulosa cell samples were collected at 24 h, 48 h, and 55 h after PMSG injection. The results showed that, compared with the control group, CORT-injected mice revealed a significant decrease in ovulation rates, ovarian weight, ovarian index, the number of secondary follicles and mature follicles, levels of estrogen and progesterone, and mRNA expression of steroid synthase-related genes. Collectively, our findings clearly demonstrated that CORT injection could represent an effective practice to simulate stresses that inhibit ovarian functions by reducing follicular development and ovulation.

## 1. Introduction

Stress is an atypical adaptive response that occurs when the body is stimulated by various internal or external environmental factors, as well as social and psychological factors [1]. Among them, “acute stress” generally shows a stress response within a few hours. If a stimulus is repeated several times a day, it can be called “subacute stress”. If stress lasts for weeks to months, it can be called “chronic stress” [2]. In conditions of modern intensive production, stressors such as restraint, weaning, transportation, acute stress, and chronic stress will cause damage to livestock and poultry [3,4]. After the initiation of stress response, the hypothalamic–pituitary–adrenal (HPA) axis is activated, and the corticotropin releasing hormone (CRH) released from the hypothalamic paraventricular nucleus enters the pituitary portal system circulation, which can upregulate the level of glucocorticoids by stimulating the secretion of adrenal gland corticosteroids (ACTH) [5,6]. Glucocorticoids are mainly consist of cortisol (in human and mammal) and corticosterone (in rodents) [7,8,9,10]. The excessive levels of cortisol/corticosterone (referred to as CORT) will disturb the internal secretion system, inhibit the immune system, impair the reproductive system, and cause reproductive failure [11,12,13]. As a by-product of the stress response, glucocorticoids have been shown to inhibit the secretion of gonadotropins in multiple species. CRH and ACTH act primarily on the hypothalamus and higher brain centers, while glucocorticoids may act on the entire hypothalamic–pituitary–gonadal (HPG) axis [14,15,16]. In the normal estrous cycle, hypothalamic secretion of gonadotropin-releasing hormone (GnRH) stimulates the release of follicle-stimulating hormone (FSH) and luteinizing hormone (LH) from the pituitary gland [17,18]. FSH and LH synergistically promote the development of ovarian follicles and the secretion of estrogen [19]. Actually, there is evidence suggesting that stress conditions will reduce the secretion of GnRH and LH in the HPA axis, leading to the blockade of ovulation [20].

At present, there are still some difficulties in directly studying the effects of stress on reproduction. This is because in modern intensive-farming environments, complex and variable stressors can cause multiple different stresses in animals [21]. Also, for specific stresses, there are different stress responses between individuals. When exploring the effects of stress on physiological processes such as reproduction, we should try to standardize repeatability, adaptability, and duration of stresses. If there are multiple sources of stress, the toxic side effects inherent in certain stressors should also be excluded. In fact, when the body is exposed to multiple stresses at the same time, the endocrine response seems to be an ideal way to coordinate such stress stimuli [22]. Studies have shown that elevated plasma levels of CORT can be used as a marker of animal stress response [2]. Therefore, we selected the final product of the HPA axis, CORT, to induce stress in the body through continuous intraperitoneal injection, and to maintain the stability and repeatability of the stress model.

It remains unclear whether CORT exerts any influence on ovarian physiology. In this study, by consecutively injecting CORT into the peritoneal cavity to form a stress state, we simulated the effects of stress on ovulation and ovarian follicular development in mice. We also explored a possible mechanism of reproductive damage under stress conditions. Our research might lay a theoretical foundation for developing technical methods to alleviate the adverse effects of various stressors on livestock and poultry reproduction.

## 2. Materials and Methods

### 2.1. Reagents and Antibodies

Phosphate-buffered saline (PBS, 20012) was purchased from Gibco (Grand Island, NY, USA). Pregnant mare serum gonadotropin (PMSG) and human chorionic gonadotropin (HCG) was purchased from Ningbo Second Hormone Factory (Ningbo, Zhejiang, China). CORT (corticosterone; HY-B1618) was purchased from MCE (New Jersey, NJ, USA). Progesterone (P4) and estradiol (E2) radioimmunoassay (RIA) test kits (B08TB, B05TB) were purchased from Beijing North Bioengineering Research Institute Co. (Beijing, China). The reverse transcriptase kit, SYBR Premix Ex TaqTM, was from TaKaRa (Tokyo, Japan).

### 2.2. Animals

All the animal experiments were performed in accordance with the guidelines of the Animal Research Institute Committee at Nanjing Agricultural University. Three-to-four-week-old female ICR (Institute of Cancer Research) mice (Qing Long Shan Co., Animal Breeding Center, Nanjing, China) were housed five per cage in a temperature-controlled (22 ± 2 °C) room with a 12:12 h light:dark cycle and had ad libitum access to water and food. The mice (three-to-four-week-old) were randomly divided into control group and CORT group. Each mouse in the treatment group was injected with CORT. CORT was dissolved in DMSO to reach a concentration 0.2 mg/μL. The control group was injected with the same volume of DMSO. The treatment procedures were briefly presented as follows: On the first day at 7:00 a.m., mice were injected with 10 IU of PMSG. The control group (n = 40) and the test group (n = 40) were injected with 5 μL of DMSO and 5 μL of CORT (0.2 mg/μL) every 8 hours. At 24 h, 48 h, 55 h after the PMSG injection, 30 mice in each group were sacrificed for collection of ovaries, granulosa cells (GCs), and blood samples. The remaining 10 mice in each group continued to be injected with 10 IU (international units) of HCG at 48 h after the PMSG injection. At 14 h after the HCG injection, oocytes were retrieved from the ampulla for calculating the number of ovulations. All blood samples were centrifuged at 2000× *g* for 20 min, and the treated serum samples were stored in a −80 ℃ refrigerator for later use.

### 2.3. Calculation of Ovulation Number 

In mice received successive injection of PMSG and hCG, the cumulus complex was collected from the ampulla of the fallopian tube, digested in hyaluronidase for 2 min, and the particles around the oocyte were removed by a thin bust pipette [23]. The number of oocytes discharged from the bilateral ovaries of one mouse was counted to assess the ovulation number.

### 2.4. Follicle Qualitative Standards

The ovaries of the mice were fixed with 4% paraformaldehyde, embedded in paraffin, serially sectioned to a thickness of 5 μm, and stained with hematoxylin-eosin (HE) [24]. The morphological characteristics of follicles at each stage can be presented as follows: (1) Primary follicle is characterized by a single layer of columnar granulosa cells is surrounding the oocyte; (2) In secondary follicles, multi-layered cubic granulosa cells is forming, beginning to secrete follicular fluid. A zona pellucida is forming around the oocyte; (3) In secondary-vesicular follicle, a small follicular cavity with a diameter of 250–450 μM is forming; (4) In mature follicle, the amount of follicular fluid and the cavity of the follicle increases, and the cumulus is forming. The follicle gradually protrudes to the surface of the ovary, and the diameter of the follicle finally reaches more than 450 μM before ovulation.

### 2.5. Radioimmunoassay

According to the instructions of the radioimmunoassay (RIA) test kit, a standard curve was drawn to calculate estradiol in serum (detection range 0.5–150 ng·L^−1^; intra-assay coefficient of variation 10%, inter-assay coefficient of variation 15%); progesterone (detection range 0.2–100 ng·mL^−1^; intra-assay coefficient of variation 10%, inter-assay coefficient of variation 15%).

### 2.6. Real-Time Quantitative Polymerase Chain Reaction (qRT-PCR)

Total RNA and cDNA were collected from granulosa cells in the right ovary of mice [25,26,27,28]. The mRNA levels of *Cyp11a1*, *Cyp17a1*, *Cyp19a1*, *Star* and *3β-Hsd* in the ovarian GCs of the tested mice were detected by real-time polymerase chain reaction (PCR). All primer sequences in the experiment were derived from NCBI (National Center for Biotechnology Information, Bethesda, MD, USA) and primers were designed by the software Primer Premier 5.0. The primer sequences of the target genes are listed in Table 1. Glyceraldehyde-3-phosphate dehydrogenase (*GAPDH*) was used as an internal control. Relative expression of target genes was analyzed by 2^−ΔΔCt^ method.

### 2.7. Statistical Analysis

All experiments were repeated at least three times, and all data were presented as means ± standard error (S.E.). Statistical significance was analyzed by the SPSS version 16.0 software (SPSS, Chicago, IL, USA). Differences between two groups were assessed using the Student *t*-test, and between multiple groups using one-way analysis of variance (ANOVA). Values of *p* < 0.05 were considered significant.

## 3. Results

### 3.1. Effect of Corticosterone (CORT) Injection on Body Weight and Ovarian Development in Mice

To assess the effects of stress models on ovarian development, we measured the gain of body weight, ovarian weight, and ovarian index at three time points. In the control group, no abnormalities were observed in the appearance and activity of the mice, and the body weight was moderately increased. Compared with the control group, the weight gain of the CORT group (1 mg/mouse) was slowed down, the action was slow, and the coat color was dull. The ovary at each stage was weighed and the organ index was calculated. The results are shown in Figure 1. At 48 h, the gain of body weight in the CORT group was significantly lower than that of the control group (*p* < 0.01). Ovarian weight and ovarian index in the CORT group was significantly lower than that of the control group (*p* < 0.05). At 55 h, the gain of body weight, ovarian weight and ovarian index of the CORT group were significantly lower than those of the control group (*p* < 0.01).

### 3.2. Injection of CORT Inhibits Ovulation in Mice

To investigate whether there was an effect on ovulation after injection of CORT (1 mg/mouse), mice (three-to-four-week-old) were injected with 10 IU of PMSG for 48 h and then injected with 10 IU of HCG. After 14 h, oocytes were collected from the ampulla of the mouse. As shown in Figure 2, ovulation occurred in both the control group and the CORT group. The number of ovulation in the control group was 53.1 ± 2.4, and the number of ovulation in the CORT group was 39.3 ± 2.0. The number of ovulation was significantly decreased after injection of CORT (*p* < 0.01).

### 3.3. Effects of CORT Injection on Mouse Follicles

Hematoxylin-eosin (HE) staining was used to investigate the effect of CORT on follicular development. The left ovaries of mice were taken at 24 h, 48 h, and 55 h respectively after PMSG injection. As shown in Figure 3, compared with the control group, CORT inhibited the development of follicles at different time periods, especially in the late stage of follicular development. At 24 h, stress had no significant effect on follicular development at each stage (*p* > 0.05); At 48 h, secondary vesicular follicles were significantly decreased (*p* < 0.01), mature follicles were significantly decreased (*p* < 0.05); At 55 h, secondary vesicular follicles and maturation follicles were significantly reduced (*p* < 0.01).

### 3.4. Effect of CORT Injection on Steroidogenesis in Mice

As shown in Table 2, the levels of progesterone and estradiol in the CORT group decreased at different time points, and progesterone decreased significantly at 24 h and 48 h (*p* < 0.05). Both progesterone and estrogen decreased significantly at 55 h (*p* < 0.01).

### 3.5. Effects of CORT Injection on the Transcription Level of Steroid Synthase Gene in Mouse Granulosa Cells (GCs)

During the synthesis of steroid hormones, the level of transcription of the hormone synthase gene can affect the development of follicles. The transcript level of steroid synthase gene in mouse GCs was detected by qRT-PCR. The results in Figure 4 showed that the transcription levels of *3β-Hsd*, *Cyp19a1* and *Star* in GCs of CORT group were significantly lower at 48 h and 55 h compared with the control group (*p* < 0.01); the transcription level of *Cyp17a1* in GCs of CORT group was significantly decreased at 48 h and 55 h (*p* < 0.05).

## 4. Discussion

In the process of intensive production of livestock and poultry, stresses are inevitable factors that affect the reproductive performance of female animals [29]. Researchers have hitherto established hundreds of animal stress models. For instance, the chronic restraint stress model is one that has been widely used. However, these models have some defects, including subjectivity, complexity in influencing factors, large variability, and being time-consuming and laborious in operation [30]. In addition, a chronic mild stress (CMS) model is also widely used in simulated stress tests. A series of stressors such as isolation, crowding, day and night reversal, forced swimming, heat stress, cold stress, fasting, and water prohibition were included in this model. However, the model is difficult to operate, requires large space and lasts for a long time. Studies have shown that in the CMS model, different strains of mice have different sensitivity to stress, resulting in large differences between individuals, and the relative severity of different stresses in the model is still uncertain [31,32]. Although the CMS model is widely used in stress experiments, it is difficult to ensure the validity of the model [33]. Nevertheless, there are to date few stress models that are specifically designed for researches in mammalian ovaries. It has been reported that hormones released by the hypothalamus during stress is maintained at a constant state, and the surge of blood CORT can be used as a marker of stress [34]. In this study, we used CORT to inject mice and successfully established an ovarian stress model. Compared with the stress models as mentioned previously, the advantages of our current model might be manifested in several aspects: 1. it provides stress with a repeatable and stable state; 2. it can be established rapidly; 3. it is easy to operate; 4. it prolongs the periods of stress state. More importantly, we demonstrated for the first time that the CORT-injection model is effective in influencing ovarian functions.

The ovarian follicles represent basic units for performing ovarian reproductive and endocrine functions. The quality and quantity of follicles determine reproductive potential and reproduction cycle [35]. In fact, studies have found that under stress, the HPG axis is affected, which will reduce the pulse secretion of GnRH, leading to decreased production of FSH, thereby blocking follicular development, resulting in reduced number of ovulations [36]. As a terminal product of the HPA axis in stress response, CORT has rarely been reported in the regulation of follicular development during stress conditions. This study demonstrated for the first time that CORT injection hindered the development of mouse follicles. Inhibition of follicular development mainly occurred in secondary follicles and mature follicles after 48 h of PMSG injection, indicating that CORT has the strongest inhibitory effect on follicles in advanced stage, especially before ovulation, leading to a decrease in the number of mature follicles and a decrease in the number of ovulations. This is consistent with the findings of ovary development in the results 1, which showed significantly reduction in the ovarian weight and ovarian index after 48 h of PMSG injection. Collectively, we demonstrated that the CORT affects ovarian and follicular development. This might provide a novel understanding into the pathogenesis of stress-regulated reproductive disorders.

There is evidence that under stress conditions, the HPA axis will reduce the pulse secretion of GnRH/LH, inhibit the formation of LH surge before ovulation, and block the ovulation [3]. Studies have shown that when the mouse chronic mild stress (CMS) model is used, it is detected that the decrease of LH level after the increase of CORT in mice leads to anovulation [23]. However, it remains unclear whether CORT has a direct effect on ovulation. In our experiment, consecutive injection of CORT caused a decrease in the number of ovulations. This result is consistent with the data in result 3, which showed impaired development of mature follicles after CORT injection. Given that administration of exogenous LH/HCG may activate the process of ovulation, we speculate that the supplement of exogenous LH/HCG will rescue the ovulation disorder caused by stress in the female. Nevertheless, further research is needed to explore how the luteinizing hormone receptor (LHR) regulates ovulation in ovarian follicles under stress conditions.

Estradiol and progesterone are important factors for follicular development [18,19]. At the same time as follicles are undergoing abnormal growth, we found that estradiol and progesterone in the experimental group decreased at various time points. The ovary in the female reproductive system produces estrogen to maintain and regulate the reproductive cycle. Previous reports have shown that estradiol plays an important role in follicular development and granulosa cell survival. When estradiol is insufficiently secreted, follicular atresia increases and follicular maturation is inhibited [37]. Although the synthesis of steroid hormones is regulated by the HPG axis, steroid-synthase-mediated biochemical reactions in ovarian granulosa cells is the basic pathway for steroid hormone synthesis [38,39]. The transcriptional levels of the hormone synthase genes can affect the amount of hormones in the follicular fluid. Genes such as *3β-Hsd*, *Cyp11a1*, *Cyp17a1*, *Cyp19a1* and *Star* are important steroid hormone synthetases. The synthesis of steroid hormones in the ovary consist of several continuous processes, including: *Star* transfers cholesterol molecules to the mitochondrial inner membrane; cholesterol molecules are converted to pregnenolone under the action of *Cyp11a1*; pregnenolone is converted to progesterone under the action of *3β-Hsd*; progesterone is converted to androstenedione by *Cyp17a1* under hydroxylation; and when androstenedione is transferred to granulosa cells, it is converted to estradiol through the action of *Cyp19a1* [40,41,42]. This shows that when the supply of progesterone in the follicle is insufficient, it also leads to a decrease in the secretion of estradiol. We measured the expression levels of these genes in the test group and the control group, respectively. Our results demonstrate that mRNA levels of *3β-Hsd*, *Cyp11a1*, *Cyp19a1*, and *Star* are significantly down-regulated after 48 h of CORT injection, consistent with the findings in Result 4, which showed a decrease in levels of hormone production after CORT injection. The results indicate that CORT interferes with the synthesis of steroid hormones by inhibiting the transcription level of the steroid synthase gene, which may be the main cause of ovulation in the stress-treated sows in the experimental group. Since CORT has been reported to affect the release of gonadotropins at the hypothalamic-pituitary level [3], it remains to be investigated whether expression changes of those steroid hormone synthetases were due to the direct effect of corticosterone on granulosa cells or through the hypothalamo-pituitary gland. Therefore, more experiments should be conducted in the future to examine the activation status of the HPG axis, including the expression of glucocorticoid receptors in the anterior pituitary gland, and FSH levels in animals that received corticosterone treatment. Nevertheless, our findings provide evidence that CORT inhibits follicular development and ovulation. Collectively, we have demonstrated for the first time that consecutive injection of CORT in mice suppressed the synthesis of important steroid hormones such as estradiol and progesterone. The decreased hormone levels might, in turn, adversely affect the development of mouse follicles, and subsequently lead to some anovulatory disorders.

## 5. Conclusions

In summary, the present study established an ovarian stress mouse model by using CORT intraperitoneal injection. The CORT treatment successfully simulates the stress response under physiological conditions, with simple operation and stable effect. At the same time, it was found that CORT can inhibit ovulation and the synthesis of ovulation-related steroid hormone in mice, suggesting that the ovulation disturbance caused by stress might be attributable to the reduction of steroid hormone secretion caused by stress. These findings might improve our understanding regarding the mechanism of stress-induced anovulation in animals, and contribute to the development of novel procedures for ameliorating the adverse effects of stress on reproductive performance in female animals.

## Figures and Tables

**Figure 1 animals-09-01047-f001:**
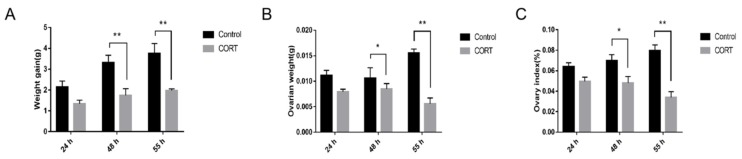
Influence of corticosterone (CORT) on body weight and ovarian development in mice. (**A**) The weight gain of mice from 0 to 24 h, 48 h, 55 h; (**B**) the total weight of the bilateral ovaries at various time points; (C) the ovaries of mice at various time points Index (ovarian index = ovarian weight/termination weight × 100%). Data are expressed as mean ± standard error (mean ± standard error (S. E.)); n = 10. * *p* < 0.05; ** *p* < 0.01.

**Figure 2 animals-09-01047-f002:**
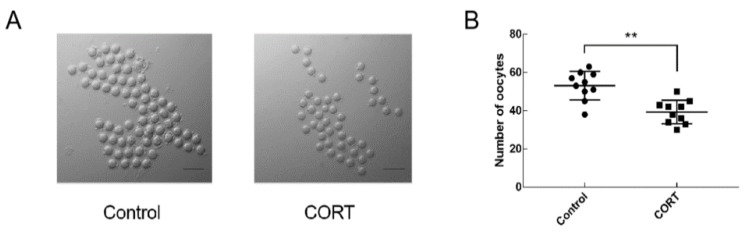
Influence of CORT on the number of ovulation in mice. (**A**) Oocyte morphology from control group and corticosterone (CORT) mice; (**B**) Quantitative statistics of the number of ovulation in each group. Data are shown as mean ± S.E.; n = 10 in each group. * *p* < 0.05; ** *p* < 0.01.

**Figure 3 animals-09-01047-f003:**
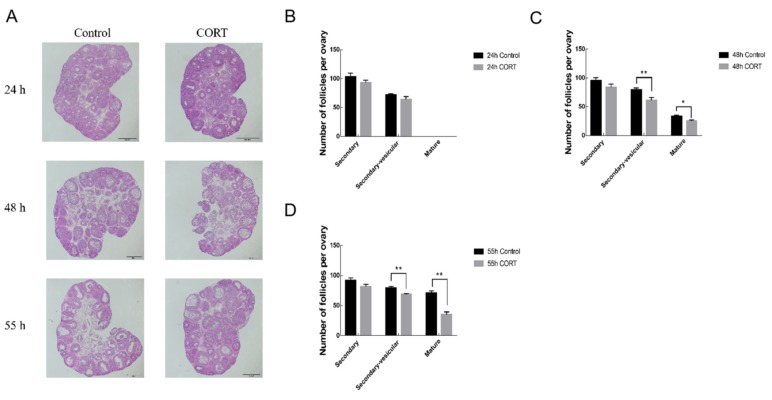
Influence of CORT on the development of mouse follicles. (**A**) Ovarian morphology of control group and CORT group at 24 h, 48 h, 55 h; quantitative statistics of secondary follicles, secondary follicles and mature follicles at (**B**–**D**) 24 h, 48 h, 55 h. Data are shown as mean ± S.E.; n = 10 in each group. * *p* < 0.05; ** *p* < 0.01.

**Figure 4 animals-09-01047-f004:**
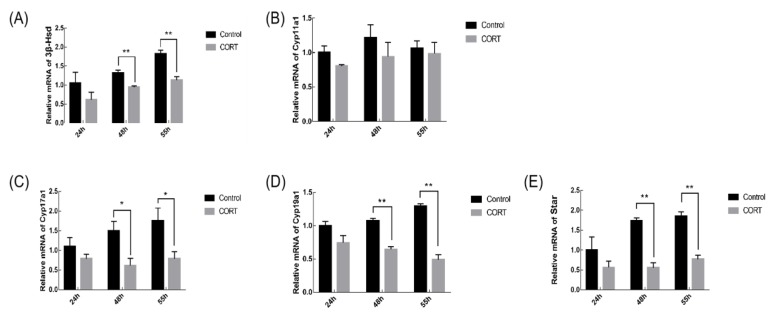
Influence of CORT on steroid synthase gene transcription levels in mouse ovarian GCs. (**A**–**E**) The expression of *3β-Hsd*, *Cyp11a1*, *Cyp17a1*, *Cyp19a1* and *Star* related genes involved in steroidogenesis in the control and CORT groups at 24 h, 48 h, and 55 h by real-time quantitative polymerase chain reaction (qRT-PCR). Data are shown as mean ± S.E.; n = 10 in each group. * *p* < 0.05; ** *p* < 0.01.

**Table 1 animals-09-01047-t001:** Primers used in the present study.

Gene	Forward Sequence	Reverse Sequence
*3β-Hsd*	5′-GCTGCACAGCCCTCCTAAG-3′	5′-TGATCCTCTGGCCCACAAAC-3′
*Cyp11a1*	5′-CCCGGAGAGCTTGTGCAAAT-3′	5′-CCCATGCTGAGCCAGATGTC-3′
*Cyp17a1*	5′-TGGAGGCCACTATCCGAGAA-3′	5′-GAAGCGCTCAGGCATAAACC-3′
*Cyp19a1*	5′-ATCCGGTTTTTAAACGGCTGC-3′	5′-TCTTGCGCTATTTGGCCTGG-3′
*Star*	5′-AACGGGGACGAAGTGCTAAG-3′	5′-CCTCTGCAGGACCTTGATCTC-3′
*Gapdh*	5′-AAGGTGGTGAAGCAGGCAT-3′	5′-GGTCCAGGGTTTCTTACTCCT -3′

**Table 2 animals-09-01047-t002:** Serum hormone levels in the experimental and control groups (mean ± S.E.)

Time (h)	Groups	Estradiol (pg/mL)	Progesterone (ng/mL)
24	Control group	51.30 ± 9.62	44.97 ± 3.86
	CORT treatment group	33.53 ± 16.46	26.50 ± 3.49 *
48	Control group	80.91 ± 7.80	31.23 ± 3.08
	CORT treatment group	57.10 ± 14.57	21.79 ± 1.96 *
55	Control group	81.70 ± 18.47	37.63 ± 1.60
	CORT treatment group	31.42 ± 5.63 **	15.33 ± 2.30 **

* *p* < 0.05; ** *p* < 0.01.

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
