# Peer review of "Corticosterone Injection Impairs Follicular Development, Ovulation and Steroidogenesis Capacity in Mice Ovary"

_animals, 2019, doi:10.3390/ani9121047_

Round 1
Reviewer 1 Report
The present study is designed to show effects of corticosterone injection on follicular development, gene expression of steroid-synthesizing enzymes, steroid hormone levels and ovulation as a model of stress condition. The authors found that corticosterone injections affected the number of ovulated oocytes, follicular development of secondary follicles into antral and preovulatory ones, and gene expressions of3bHsd, Cyp17a1, Cyp19a1 and Star. Based upon the data presented in the manuscript, it is not clear whether those changes were due to the direct effect of corticosterone on granulosa cells or through the hypothalamo-pituitary gland. Discussion on this issue may be necessary. Expression of glucocorticoid receptors in the anterior pituitary gland in mice may be examined. FSH levels may be studied in corticosterone-treated animals.
Lines 85 and 139: The age of animals given hormone treatment should be shown.
Lines 87 and 88: Preparation of corticosterone should be explained briefly. Solvent of corticosterone is DMSO alone?
Line 105: The term “foam” is not appropriate.
Line 115: Gene names of CYP11a1, CYP17A1, CYP19A1, StAR and 3b-HSD are not correct. Those mouse genes are described as Cyp11a1 and so on.
Line 150: The term “cavity”may be “vesicular”.
Line 151: “secondary cavities Bot follicle” should be revised properly.
Line 157: “Figure 3” should be “Table 2”.
Line 229: Citation [37] is not correct.
Reviewer 2 Report
animals-641957
The manuscript titled Corticosterone injection impairs follicular development, ovulation and steroidgenesis capacity in mice ovary by Wei and co-workers investigated intraperitoneal injection of CORT three times in a 24 hr period with mice collected at 24, 48, and 55 hr to determine influence on follicular development and steroidogenesis. There are several things that are troubling regarding this manuscript. First off the authors have stated there were 80 animals in the experiment separated into two groups (CORT and Control) which suggests 40 animals per treatment. Yet, number of animals collected at each time point per treatment is 10—with only three collection periods this leaves 20 animals unaccounted for. Secondly the drug that was used for the treatment is referred to only as CORT throughout the manuscript. The catalog and manufacturer is listed in the methods section, but how many readers would go to the trouble to look this up. With the number of glucocorticoids available, I feel being clear and specific in the manuscript is imperative. The measurable weight differences among groups (Figure 1, A) is concerning. It is not clear if this is due to a direct effect of treatment or a consequence of food intake. It is also concerning that the repeated injections are stressing the mice. How is handling stress differentiated from the exogenous CORT? Since the HPG axis was not assessed in these experiments, it cannot be concluded that treatment affected steroid synthesis directly. This should be clarified in the conclusions. Finally, there are language use errors throughout the manuscript that need to be cleaned up before the manuscript is acceptable for publication.
Other considerations:
Line 88. Mice were injected 7:00, 15:00, and 23:00 daily. When was the injections started relative to PMSG treatment? Specify how many days they were treated. This is also confusing since several times in the manuscript this treatment is referred to as “continuous injection” (eg. Line 248). Is it continuous or is it repeated?
Line 146. Write out HE since it is at the beginning of the sentence.
Line 151. The 55 hr result is incomplete and confusing. Reword for clarity.
Line 154 – 157. This is discussion and should be removed from the results section.
Line 157. Concentrations of steroids are not shown in Figure 3, as stated. These results are in Table 2. This needs to be corrected.
Line 158 – 159. Punctuation seems incorrect making the sentence confusing. Correct for clarity.
Line 167. P > 0.05 is not a trend for difference—P > 0.05 is in fact indicates a lack of difference.
Line 187. Can rather than could
Line 188. “Long time” is subjective—please clarify.
Throughout the discussion the results are referred to (Result X). The specific reference to the result is not necessary within the discussion.
Line 203. CORT was administered exogenously in this study so the effect on the HPA axis was not measured. HRA seems to be a typographical error.
Line 201. It is not clear what “development of the ovary” refers to. Is this weight? Development was not assessed.
Line 207. Peak, I feel should be referred to as “surge”
Line 208. CMS should be spelled out. I am uncertain what this is referring to.
Line 210. Citation 23 did not include any data regarding LH or ovulation
Line 224. “dysplasia” is the wrong word as it refers to abnormal growth. There is not really abnormal growth in the treated mice.
Line 231. Spell out GCs
Figure Legends should stand alone and not draw conclusions. The reader should make their own conclusions regarding the data. Suggest “Influence of CORT on body weight”. Rather than “Injection of CORT affect body weight”.
Line 392. “Ovarian” rather than “follicle” morphology.
Round 2
Reviewer 1 Report
The manuscript has been properly revised according to the comment.
I found the one typying error in Line 38 (initiayon).